# Ginsenoside Re as a Probe for Evaluating the Catalytic Potential of Microcrystalline Cellulose for the Degradation of Active Pharmaceutical Ingredients

**DOI:** 10.3390/ph18060869

**Published:** 2025-06-11

**Authors:** Xinyu Gao, Shengyuan Xiao

**Affiliations:** Engineering Center of Chinese Ministry of Education for Edible and Medicinal Fungi, Jilin Agricultural University, Changchun 130118, China; 13188538499@163.com

**Keywords:** microcrystalline cellulose, catalytic potential, active pharmaceutical ingredient, degradation, evaluation method

## Abstract

**Background/Objectives:** Microcrystalline cellulose (MCC) is a commonly used pharmaceutical excipient. At present, the catalytic potential of MCCs for the degradation of active pharmaceutical ingredients (APIs) has not been paid adequate attention. This study aims to investigate the representativeness of the pH value of an MCC determined in accordance with the pharmacopeia method to the acidity on its surface. **Methods:** We tested the differences between the catalytic activities of different MCCs and their supernatant prepared in accordance with the pharmacopeia method for the hydrolysis of ginsenoside Re, which is relatively stable in neutral or weak alkaline aqueous solutions but sensitive to acid. The sulfur content of the sulfuric acid-prepared MCC was measured using an ICP-OES. **Results:** All of the five tested commercially available and two self-prepared MCCs have been found to significantly promote the hydrolysis of ginsenoside Re. But their supernatants were neutral and chemically inert to Re. The sulfur content of the MCC prepared in this experiment using sulfuric acid hydrolysis was determined to be 109.60 µg/g, which is equivalent to 186 to 465 µM of sulfuric acid on the surface. **Conclusions:** The pH value of an MCC determined in accordance with the pharmacopeia method is not representative of the acidity on its surface. The primary reason should be that there is immobilized acid that is not so easily dissociated into the media. Ginsenoside Re is sensitive and applicable as a probe for the evaluation of the catalytic potential of pharmaceutically used MCCs.

## 1. Introduction

Microcrystalline cellulose (MCC) is a commonly used pharmaceutical excipient, especially as a tableting excipient, due to its excellent compressibility properties [1,2].

Although there have been many technologies developed for MCC preparation [3,4], MCCs for pharmaceutical use shall be prepared from native cellulose using mineral acids according to the pharmacopeia of different regions or countries [5,6,7].

The characterization of an MCC for pharmaceutical use is usually based on its physiochemical properties, e.g., organic impurities, pH, polymerization degree, content of crystalline domains, particle size, density and compressibility index, angle of repose, powder porosity, hydration and swelling capacity, moisture sorption capacity, and moisture content [4,7,8]. The potential of an MCC for the degradation of active pharmaceutical ingredients (APIs) has not hitherto attracted adequate attention due to the common knowledge that MCC is chemically inert to APIs. Among the characteristics mentioned above, the pH value is the only factor that might be related to the catalytic activity for the degradation of APIs. The moisture sorption capacity and moisture content are more related to the tabletting technology. Mihranyan A. et al. [9] reported that MCC promotes the hydrolysis of moisture-sensitive drugs such as aspirin (ASA). Commercially available MCCs have been reported to promote the degradation of perindopril erbumine (PER) and enalapril maleate (EM) in binary mixtures [10]. They suggested that the water-binding energy is the main factor associated with the stability of APIs in binary mixtures with MCCs [9,10,11,12]. However, Carlsson et al. reported that the surface charge (of carboxyl-functionalised and neutralized MCC) is the determining factor for ASA hydrolysis [13]. The surface charge is an indirect characteristic of the pH value; it is not applicable when the acid has been neutralized. Obviously, free water is much less powerful than acid for ASA hydrolysis. There has been no other report about the effects of MCCs on the degradation of APIs [7,14].

It has been noticed that some natural compounds, which are stable in neutral aqueous solution but acid-sensitive, are hydrolyzed under the catalysis of several commercially available MCCs. The pH values determined in accordance with the pharmacopeia methods were observed in this study to be inconsistent with the hydrolysis catalytic activity. That is, the acidity of the supernatant is not so representative of the acidity on the surface of the MCC. Methods by which to characterize MCCs from the point of view of the stability of the APIs are thus urgently needed [10].

In this study, we report the differences between the catalytic activities of different MCCs and their supernatant prepared in accordance with the pharmacopeia method for the hydrolysis of ginsenoside Re, which is relatively stable in neutral or weak alkaline aqueous solution but sensitive to acid, and the feasibility of ginsenoside Re as a probe to evaluate the catalytic potential of pharmaceutically used MCCs on APIs based on this reaction.

## 2. Results and Discussion

### 2.1. Ginsenoside Re Hydrolysis Catalyzed by Diluted Acid and Microcrystalline Cellulose

Figure 1 shows representative total ion chromatograms (TICs) of ginsenoside Re and the transformed products under the treatment of dilute acid, sodium carbonate, and microcrystalline celluloses. Ginsenoside Re was observed to be quite stable in neutral aqueous solution (Figure 1B) but unstable in acidic solution (Figure 1C). The glycosidic bond at C-20 of Re has been hydrolyzed by very dilute sulfuric acid to produce 20(R/S)-ginsenoside Rg2, while it was observed to be quite stable in dilute alkaline solution (Figure 1D). On the other hand, ginsenoside Ro was observed to be quite stable in dilute acidic solution (Figure 1C) but unstable in alkaline solution (Figure 1D). Approximately 60% of the Ro has hydrolyzed to produce zingibroside R1 (zR1) after 6 h of incubation (90 °C) in 1.0% sodium carbonate solution.

Ginsenoside Re was observed to have been hydrolyzed by commercially available MCCs (Figure 1E,F). The determination of ginsenoside Re and Rg2 were described in the Appendix A (Refs [15,16,17] are cited in the Appendix A). The hydrolysis rates of the two MCCs were quite different from each other, but the pH values of their supernatants were determined to be almost the same. One of the commercially available MCCs displayed catalytic activity for Re hydrolysis close to that by 9.2 μM sulfuric acid (Figure 1C,E). According to a previous report [18], the activation energy of perindopril erbumine increased significantly when it was absorbed on the fibre of an MCC; there must be a catalytic factor on the surface of the MCC—other than the free water [11] and surface charge [13]—that would make it much more powerful than 9.2 μM sulfuric acid for the acidic hydrolysis of Re.

### 2.2. Hydrolysis Catalytic Activities of Acid-Treated Cellulose for Ginsenoside Re Hydrolysis

To investigate the determining factors of Re hydrolysis, the hydrolysis catalytic effects of cotton wool, filter paper, silicon, and acid-treated cotton wool have been tested. Ginsenoside Re was observed to be stable in neutral solution under the conditions of this experiment (Figure 2A). The native celluloses, cotton wool, and filter paper also did not show significant catalytic activity. Silicon is known to have weak acidity, but its hydrolysis catalytic activity on Re was not observed (similar to the cotton). The cotton wool treated with 2N sulfuric acid or hydrochloric acid for different periods of time of incubation, as well as that without heating, showed very significant hydrolysis catalytic activity on Re (Figure 2B,C). To neutralize the catalytic activity of the acid-treated cellulose, it was dispersed in 1.0% (*w*/*v*) sodium carbonate solution and immersed overnight after 10 min of transonic treatment in order to help the alkaline solution penetrate the cellulose. The celluloses were rinsed with 5000 volumes of distilled water. The hydrolysis of Re was diminished after the neutralized MCCs (Figure 2E,F).

Ginsenoside Re is not moisture-sensitive. All the reactions in this study were performed in aqueous solution, so the free water content is not the main factor for the hydrolysis. Cotton wool that was immersed in acidic solution without heating and then rinsed to neutrality displayed similar catalytic activity to those that were incubated for different periods of time. This observation indicates that the acidic residue attached on native, partial hydrolyzed celluloses, and MCCs and suggests that the acidic residue is a dominant factor in the Re hydrolysis. This suggestion was further proven by the observation that the catalytic activity of MCCs was diminished when the acid-treated cellulose was neutralized using alkali. The transformation rates of Re decreased to 4.45% (Table 1, MCC-Sn1) and 1.17% (Table 1, MCC-Cn1), respectively, after the MCCs were neutralized using 1.0% Na_2_CO_3_ solution.

All the supernatants from the cellulose were determined to be neutral after the celluloses were eluted using 200 volumes of water. The supernatants indeed showed much weaker hydrolysis catalytic activities on Re (Figure 2D; Table 1) than the acid-treated celluloses. The MCCs prepared using both sulfuric acid (MCC-S) and hydrochloric acid (MCC-C) displayed considerable acidic hydrolysis catalytic activities, but their supernatants were all neutral. These results indicate that the acidic residues have not been dissociated from the fibre after rinsing; thus, they are not easily dissociated into the medium using the pharmacopeia method.

It is reported that sulfuric acid can form a half sulfuric ester with the hydroxyl groups on MCC saccharides [19]. For an intuitive estimation of the acidity on the surface of a swollen MCC, the sulfur content was equated to the amount of sulfuric acid. The concentration of the sulfuric acid on the surface of an MCC can be calculated based on the sulfur content against the swollen volume of the MCC. The acidity on the surface of a swollen MCC was equivalent to that of a 17–85 mM sulfuric acid solution (*v*/*v*) based on data reported by Marchessault et al. [19]. The sulfur content of the MCC prepared in this experiment using sulfuric acid hydrolysis was determined to be 109.60 µg/g, which is equivalent to 186 to 465 µM of sulfuric acid on the surface of the MCC according to the swollen rate. This type of immobilized acid cannot easily be dissociated into the supernatant. The immobilization of half acidic ester might also occur when MCCs are prepared using phosphoric acid [20] and other polybasic acids. The sulfur contents of MCC1, MCC2, and the cotton wool were 4.76, 19.66, and 43.97 µg/g respectively. However, these results are not consistent with the hydrolysis catalytic activities of the MCCs (see below). These results might be because the sulfur was not in the half sulfuric ester form in those MCCs or cotton wool.

Theoretically, hydrogen ions can contact the fibres of MCC through hydrogen bonding. This interaction might also lead to the higher acidity on the surface of an MCC than in its supernatant [13]. The acidic residue that remained in the fibre of the hydrochloric acid made MCC in this study might be due to the hydrogen bonding of H_3_O^+^ with the hydroxyl groups of the cellulose.

Although the mandatory limits for the pH value of an MCC supernatant are provided in most national or regional pharmacopeias, the supernatant pH value of an MCC would not represent the acidity on its surface according to the above observations. The binding ions on the surface of a solid are usually measured through the ζ-potential [13], but it is not applicable to determine the pH of a neutralized MCC. Ginsenoside Re is water-soluble and stable enough in neutral aqueous solution and very acid-sensitive. It might be a suitable probe for the evaluation of the catalytic potential of an MCC for active pharmaceutical ingredient acidic hydrolysis.

pH indicators are commonly used for acidity tests, and theoretically, they can be used as molecular probes to detect the acidity on the surface of an MCC. The colour development of the most acidic MCCs (Figure 1E) by methyl red, bromocresol green, and thymol blue has been attempted, but the change in colour cannot be easily observed with the naked eye. Whether it can be detected using a reflective UV or IR detector has not been determined in this study.

### 2.3. Sensitivity of Ginsenoside Re to Dilute Acid

The hydrolysis rates of ginsenoside Re by different concentrations of sulfuric acid under the conditions of this experiment are shown in Table 2. Ginsenoside Re was observed to be very sensitive to dilute sulfuric acid; it can be almost totally hydrolyzed by 14.72 µM sulfuric acid and produces mainly 20(R/S)-ginsenoside Rg2. Approximately 8% of the Re is hydrolyzed by 3.68 µM aqueous sulfuric solution at 90 °C in 6 h. The repeatability of these reactions is good according to the relative standard derivation observed.

### 2.4. Effects of Commercial Microcrystalline Cellulose on Ginsenoside Re Hydrolysis

The hydrolysis catalytic activities of 5 commercially available MCCs for the hydrolysis of Re are shown in Table 3. The pH value of the deionized water used was 6.70 ± 0.03, and those of the supernatants of all MCCs were between 6.46 and 7.10.

The data in Table 3 indicate that the acidic catalytic activities of all tested MCCs are higher than those of their supernatants. MCC-1 and MCC-2 displayed considerable acidities, while the others showed slight acidic hydrolysis catalytic activities. However, their supernatants showed similar hydrolysis catalytic activities for Re hydrolysis. These results support the conclusion that the acidity of the supernatant of an MCC does not represent the acidity on the surface.

In accordance with the various pharmacopeias, MCCs for pharmaceutical use shall be prepared using a mineral acid. MCCs are generally produced by a procedure of acidic hydrolysis, washing, bleaching, and drying. There is no alkaline substance introduced during the process, but an alkali is sometimes introduced to neutralize the acid during the procedure [8,21]. Just as H_3_O^+^ remains in the fibres, alkali might also remain. As we described in Section 2.1 (Figure 1D), ginsenoside Ro is alkaline-sensitive and could serve as a probe to evaluate the alkaline catalytic potential of an MCC. The sensitivity of the Ro to dilute sodium carbonate is shown in Table 4.

## 3. Materials and Methods

### 3.1. Materials

HPLC-grade methanol and acetonitrile were purchased from Merck (Darmstadt, Germany). Microcrystalline cellulose 1 (MCC-1, for chromatography separation use, series No. 68005761; 20–100 μm; 6% moisture; pH 7 for 10% supernatant) was purchased from Sinopharm Chemical Reagent Co., Ltd. Shanghai, China. Microcrystalline cellulose 2 (MCC-2, for R&D use, series No. C104841, Lot No. i1825025; 90 μm; 2.55% moisture; pH 5.82 for 10% supernatant) was purchased from Aladdin Industrial Cooperation, Shanghai, China. Microcrystalline cellulose 3 (MCC-3, for food or pharmaceutical use, Lot No. 20181219, PH101, complies with Chinese national standard GB 1886.103-2015 [22]) was purchased from Henan Wan Bang Industrial Co., Ltd., Zhengzhou, Henan, China. Microcrystalline cellulose 4 (MCC-4, for food or pharmaceutical use, Lot No. 20190219, PH102, complies with Chinese Pharmacopeia 2015) was purchased from Nanjin Na Man Biological Science and Technology Co., Ltd., Nanjin, Jiangsu, China. Microcrystalline cellulose 5 (MCC-5, for food or pharmaceutical use, Lot No. 170618, PH101, complies with Chinese national standard GB 1886.103-2015 [22]) was purchased from Qu Fu Medicinal Supplements Co., Ltd., Qufu, Shandong, China. Ginsenoside Re, Rg2, Ro, and zingiberoside R1 were purchased from Chemfaces (Wuhan, China). The purities of all of the compounds were higher than 98%.

### 3.2. Methods

#### 3.2.1. Preparation of Microcrystalline Cellulose and Partially Hydrolyzed Cellulose

Approximately 2 g torn cotton wool and 30 mL 2N sulfuric acid or hydrochloric acid was placed in each 50 mL flask. One group of the flasks were left without heating (0 h), others were incubated in an 80 °C water bath for 0.5 h, 1 h, and 4 h; then, the cellulose in the flask was taken out and filled tightly into a glass column. The column was then eluted using 200 volumes of deionized water, and the cellulose was drained by reduced pressure after the pH of the aqueous eluent was tested to be neutral and then rinsed using 50 mL 95% ethanol. The rinsed cellulose was finally drained and taken out from the column and air-dried. The cotton wool treated for 0 h was cellulose that had been immersed sufficiently in the acid solution and then rinsed to neutrality without the water bath incubation. The microcrystalline celluloses prepared using sulfuric acid (MCC-S) or hydrochloric acid (MCC-C) with 4 h incubation is very fine powder. The sizes of the particles were estimated under a microscope (xsp-02 medical microscope, Mofi Electronic Accessories Co., Ltd., Fuzhou, Fujian, China).

#### 3.2.2. Determination of pH of Microcrystalline Cellulose

The pH value of the MCCs was tested in accordance with the method described in USP40 NF35 [6]. Briefly, 5 g of an MCC was suspended in 40 mL of distilled water for 20 min. The pH value was estimated using a desktop pH meter (PHS-25, Inesa Scientific Instrument Co., Ltd., Shanghai, China) at room temperature.

#### 3.2.3. Neutralization of the Acidic Residue of the Microcrystalline Cellulose

The dried MCC was suspended in 1.0% sodium carbonate solution (*w*/*v*), and the suspension was treated in an ultrasonic extractor for 10 min and then left overnight. The swollen MCC was tightly filled into a column; then, the column was eluted with 5000 (or 15,000) volumes of deionized water. The MCC was finally taken out and air-dried after the column was drained and rinsed using 95% ethanol. The neutralized MCCs were named after the prepared method. MCC-Sn1 represents the microcrystalline cellulose made using sulfuric acid that was neutralized using sodium carbonate prior to rinsing using 5000 volumes of water; MCC-Cn1 represents the microcrystalline cellulose made using hydrochloride acid that was neutralized using sodium carbonate prior to rinsing using 5000 volumes of water.

#### 3.2.4. Determination of Sulfur in Microcrystalline Cellulose

The sulfur content of an MCC was measured with the assistance of the Analytical Center, Tsinghua University, using an ICP-OES (iCAP 6300, ThermoFisher, Waltham, MA, USA). Briefly, approximately 200 mg cellulose and 6 mL electronic-grade nitric acid were mixed in a closed vessel of a microwave-assisted digestion device (MARS 6, CEM, Matthews, NC, USA) and kept at ambient temperature overnight. Then, the sample was digested using a programmed microwave-assisted method, and the digested solution was diluted to 25 mL for the sulfur determination. The detector responses were recorded at 180.7 nm.

#### 3.2.5. Hydrolysis Catalytic Activity of Cellulose

The cellulose (approximately 200 mg), ginsenoside Re solution and pure water were put into a 10 mL glass vial and mixed well with vortexing. The vial was sealed and incubated in a 90 °C water bath for 6 h, and the same volume of methanol was added into the vial after it was cooled under tap water. The suspension was mixed for 5 min by vortexing then taken out after 30 min and centrifuged for 5 min at 10,000 rpm before the analysis. The same volume of ginsenoside Re aqueous solution was incubated in the water bath for 6 h as a control.

#### 3.2.6. Hydrolysis Catalytic Activity of Cellulose Supernatant

The suspension of an MCC was prepared in accordance with the method described in USP40 NF35 [6]. Nine hundred microlitres of supernatant and 100 µL of Re solution were added to a 10 mL glass vial that was sealed and incubated in a 90 °C water bath for 6 h, and the reactant was treated by the same procedure as described in Section 3.2.5.

#### 3.2.7. Determination of Ginsenosides and Their Hydrolysates

The analyses were performed using an AB SCIEX 4500 LC MS/MS (SCIEX, Marlborough, MA, USA) system consisting of a TripleQuad MS spectrometer and a SCIEX ExionLC system that includes a degasser, an auto-sampler, 2 pumps, a column oven, and a controller. The separation was performed using a Chromolith eRP C18 silicon column (50 × 2 mm, Merck, Germany). The mobile phase consisted of 3.0 mM ammonium hydroxide aqueous solution (A) and acetonitrile (B) in a gradient eluting programme. The gradient programme was as follows: 10% B was maintained for the first 2 min, followed by a linear gradient of 10–40% B for 2–5 min, 40–50% B for 5–8 min, 50–70% B for 8–9 min, and 70–95% B for 9–12 min. The flow rate was 0.3 mL/min. The determination of ginsenosides was performed in negative MRM mode by monitoring the ion transmissions at 945.5/945.5 (DP, 120 v and CE 20 v), 945.5/783.4 (DP, 120 v and CE 35 v) for ginsenoside Re; 783.4/783.4 (DP, 120 v and CE 20 v), 783.4/637.4 (DP, 120 v and CE 36 v) for ginsenoside Rg2; 955.5/955.5 (DP, 120 v and CE 20 v), 955.5/793.4 (DP, 120 v and CE 60 v) for ginsenoside Ro; and 793.4/793.4 (DP, 120 v and CE 20 v), 793.4/631.4 (DP, 120 v and CE 64 v) for zingibroside R1. The ion source parameters were as follows: curtain gas, 35 psi; ion spray voltage, −4500 V; gas 1, 50 psi; gas 2, 60 psi; temperature of gas 2, 650 °C. The interface parameters were as follows: declustering potential, −120 V; enhancement potential, −10 V; collision cell exit potential, −15 V.

#### 3.2.8. Evaluation of the Catalytic Potential of Commercially Available MCCs for Ginsenoside Re Hydrolysis

The reactions of the MCCs (or their supernatants) catalyzed ginsenoside Re hydrolysis were performed using the same procedure as described in Section 3.2.5 and Section 3.2.6. The percentage of Rg2 against the total amount of Re and Rg2 in a hydrolysate was used to represent the hydrolysis rate of ginsenoside Re under the catalysis of an MCC.

## 4. Conclusions

Our observations indicated that the remaining acidic residue is the dominant factor governing the catalytic potential of microcrystalline cellulose for the degradation of pharmaceutical active ingredients. The pH value of the supernatant of microcrystalline cellulose does not represent the acidity on its surface. The first reason is because there might be immobilized acid that is not so easily dissociated into the media, and the second is that hydrogen bonding could make the H_3_O^+^ content on the surface richer than that in the supernatant. The effect of an MCC on the stability of APIs should be evaluated, alongside pH testing, when it is prepared or selected as a pharmaceutical excipient. Ginsenoside Re is sensitive and applicable as a probe for the evaluation of the catalytic potential of pharmaceutically used MCCs for the degradation of pharmaceutical active ingredients.

## Figures and Tables

**Figure 1 pharmaceuticals-18-00869-f001:**
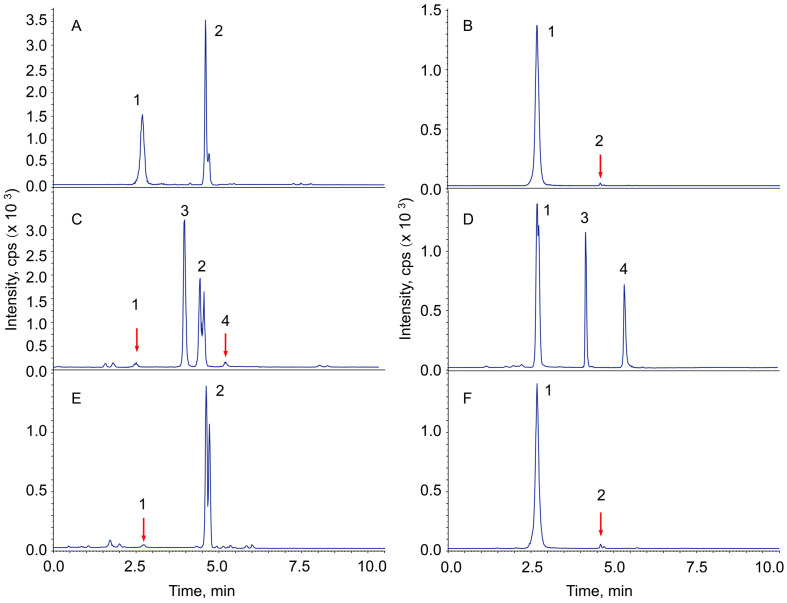
Representative total ion chromatograms of ginsenoside Re, Ro and their transformed products using alkali, acid and MCCs, The chromatograms (TICs) are the sum of the detector responses of all the ions listed in the Appendix A. (**A**) Chromatogram of the reference materials. (**B**) Control, Re aqueous solution incubated in a 90 °C water bath for 6 h. (**C**) Ginsenoside Re transformed products in 0.5 ppm sulfuric acid (*v*/*v*, 9.2 μM, Ro was added as a reference). (**D**) Ginsenoside Re transformed products in 0.5 ppm sodium carbonate (*w*/*v*, Ro was added as a reference). (**E**,**F**) Ginsenoside Re transformed products catalyzed by commercially available microcrystalline celluloses: 1, Re; 2, 20(R/S)-Rg2; 3, Ro; 4, zingibroside R1.

**Figure 2 pharmaceuticals-18-00869-f002:**
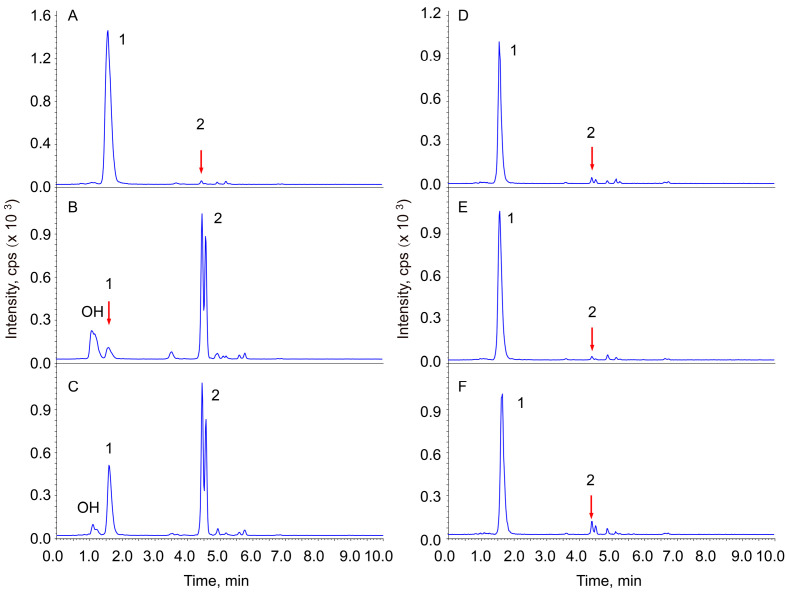
Effect of native or partially hydrolyzed cellulose on ginsenoside Re hydrolysis. The chromatograms are the TICs: (**A**) control; (**B**) cotton wool immersed in HCl without heating incubation (0 h); (**C**) MCC-C; (**D**) supernatant of MCC-C; (**E**) MCC-Cn1; (**F**) MCC-Sn1. 1, Re; 2, 20(R/S)-Rg2. OH represents a group of products of Re produced from an additional reaction at the Δ (23, 24) double bond with H_2_O.

**Table 1 pharmaceuticals-18-00869-t001:** Effects of MCCs and their supernatants on the hydrolysis of ginsenoside Re.

MCC	Re Hydrolysis Rate %	
	MCC	Supernatant
Ctrl	-	1.5 ± 0.5
MCC-S	89 ± 10	2.8 ± 0.3
MCC-C	71 ± 10	1.9 ± 0.2
MCC-Sn 1	4.4 ± 0.7	-
MCC-Cn 1	1.2 ± 0.2	-

Ctrl, control (neutral Re aqueous solution incubated in 90 °C water bath for 6 h); MCC-S, microcrystalline cellulose prepared using sulfuric acid; MCC-C, microcrystalline cellulose prepared using hydrochloric acid; MCC-Sn 1 and MCC-Cn 1 represent the neutralized MCCs; “-” means not available or unsuitable.

**Table 2 pharmaceuticals-18-00869-t002:** Sensitivity of ginsenoside Re hydrolysis to different concentrations of sulfuric acid.

Concentration ^a^, μM	Calculated pH	Re Hydrolysis Rate, %	RSD, %
Ctrl	-	1.5 ± 0.5	35.6
14.72	4.5	99.7 ± 0.1	0.1
7.36	4.8	96.9 ± 0.7	3.7
3.68	5.1	8.5 ± 1.3	15.0

^a^ The concentration of sulfuric acid was the final concentration in the reactant mixture. The hydrolysis rate was described as $\bar{x}\pm STD$ (*n* = 3).

**Table 3 pharmaceuticals-18-00869-t003:** Effects of commercial MCCs on the hydrolysis of ginsenoside Re.

MCC	Re Hydrolysis Rate %	
	MCC	Supernatant
Ctrl	-	1.5 ± 0.5
MCC-1	80.2 ± 10.7	3.3 ± 0.0
MCC-2	52.1 ± 0.3	0.6 ± 0.1
MCC-3	4.7 ± 0.0	0.5 ± 0.0
MCC-4	5.3 ± 0.6	0.7 ± 0.0
MCC-5	9.9 ± 0.1	0.3 ± 0.0

The hydrolysis rate was described as $\bar{x}\pm STD$ (*n* = 3).

**Table 4 pharmaceuticals-18-00869-t004:** Sensitivity of ginsenoside Ro to acidic and alkaline conditions.

	Concentration, ppm	Calculated pH	Ro hydrolysis Rate %
Ctrl	**-**	-	6.1 ± 1.0
Sulfuric acid	0.8 ^a^	4.5	6.3 ± 1.3
	0.4	4.8	11.6 ± 3.0
	0.2	5.1	5.0 ± 0.0
Na_2_CO_3_	80 ^b^	10.5	85.9 ± 13.3
	0.8	9.0	37.4 ± 0.4
	0.4	8.7	22.1 ± 4.7

^a^ The concentration of sulfuric acid was the final concentration of the acid by *v*/*v*. ^b^ The concentration of sodium carbonate was the final concentration of the alkali by *w*/*v*. The percentage of zR1 against the total amount of Ro and zR1 in a hydrolysate was used to represent the hydrolysis rate of ginsenoside Ro. The detail of the determination of ginsenoside Ro and zingibroside R1 was described in the Appendix A. The hydrolysis rate was described as $\bar{x}\pm STD$ (*n* = 3).

## Data Availability

Data are contained within the article and Appendix A.

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
