# Peer review of "Ginsenoside Re as a Probe for Evaluating the Catalytic Potential of Microcrystalline Cellulose for the Degradation of Active Pharmaceutical Ingredients"

_pharmaceuticals, 2025, doi:10.3390/ph18060869_

Round 1
Reviewer 1 Report
Comments and Suggestions for Authors
The article seems short but comprehensive, and the aim of this study (Evaluating the Catalytic Potential of Microcrystalline Cellulose for the Degradation of Active Pharmaceutical Ingredients, focusing on ginsenoside Re) is interesting. Some minor revision needs to be done before accepting the manuscript!
- Do not use the word we, try to use it has been noticed or it has been shown.
- Some conclusions are mentioned in the introduction in the last but one paragraph, and it is not clear if there are conclusions from this article or from previously published articles. The last two paragraphs should be merged!
- Line 69, the symbol is altered!
- For the cellulose type, the manufacturer, country, state, and city should be mentioned.
- Line 97 (microscope - manufacturer, country, state, and city) and evaluation conditions.
- The pH evaluation of the type of pH-meter used. At what temperature was this assay done?
- What amount/volume in mL of water has been used? The term volumes is very vague.
- Please mention the purity of acetonitrile!
- Table 1 should be moved to appear after the title!
- How was the cellulose type selected?
Author Response
Comments and Suggestions for Authors
Reviewer 1:
Under the Editor’s requirements, the Abstract has been revised to consist of five sections with headings of Background/Objectives, Methods, Results, and Conclusions. And the structure of the manuscript has been changed to the following sections: Introduction, Results, Discussion, Materials and Methods, and Conclusions.
The article seems short but comprehensive, and the aim of this study (Evaluating the Catalytic Potential of Microcrystalline Cellulose for the Degradation of Active Pharmaceutical Ingredients, focusing on ginsenoside Re) is interesting. Some minor revision needs to be done before accepting the manuscript!
Comment 1: Do not use the word we, try to use it has been noticed or it has been shown.
The sentence on line 52 has been revised according to your suggestion (line 57 in the revision).
Comment 2: Some conclusions are mentioned in the introduction in the last but one paragraph, and it is not clear if there are conclusions from this article or from previously published articles. The last two paragraphs should be merged!
The description of “As described in Section 3.1.2 (line 292)” has been revised to “As we described in Section 2.1 (Fig. 1d)”. The first sentence in Section 4, “According to the results of this study”, has been revised to “Our results indicated that”
Comment 3: Line 69, the symbol is altered!
The symbol in line 69 is because the font type was changed automatically when the text was rearranged according to the template. The correct font is “Symbol”. Similar errors at lines 71, 165, 184, 188, and 257 have also been corrected.
Comment 4: For the cellulose type, the manufacturer, country, state, and city should be mentioned.
The MCC type and the manufacturer were described in the Materials Section.
Comment 5: Line 97 (microscope - manufacturer, country, state, and city) and evaluation conditions.
The type and the manufacturer of the microscope have been added.
Comment 6: The pH evaluation of the type of pH-meter used. At what temperature was this assay done?
The information has been added.
Comment 7: What amount/volume in mL of water has been used? The term volumes is very vague.
The sample preparation information about the MCC pH testing has been added.
Comment 8: Please mention the purity of acetonitrile!
The purity of the acetonitrile is HPLC grade. This information was described in the Materials Section.
Comment 9: Table 1 should be moved to appear after the title!
Table 1 has been moved behind Fig. 1.
Comment 10: How was the cellulose type selected?
We tested all the commercially available MCCs in Changchun, China.
Reviewer 2 Report
Comments and Suggestions for Authors
The subject matter of the article is undoubtedly relevant and merits further discussion. Microcrystalline cellulose (MCC) is a prevalent constituent in pharmaceutical formulations; however, the extent of its potential interactions with active pharmaceutical ingredients (API) remains insufficiently characterised. In the context of increasing regulatory interest in the stability of dosage forms this topic is of clear practical importance.The submitted manuscript is satisfactory in general and appropriate for publication in the journal. Although the experimental data appear to support the hypothesis that MCC may contribute to the degradation of Re under the specified conditions, there are several significant and minor inaccuracies that must be rectified before the manuscript can be considered for publication.
Major Comments:
The selection of ginsenoside Re as the model compound is a contentious issue. The structural complexity of Re, in conjunction with the absence of a well-defined and characterised degradation profile, introduces ambiguity. The isolation and structural confirmation of the degradation products was not carried out, which limits the interpretability of the results. A more suitable choice would be a well-characterised, pharmaceutically relevant compound with known degradation pathways.
The experimental setup involves exposure to elevated temperatures (a water bath at 90°C for 6 hours), which are known to promote the degradation of many natural compounds. While this approach may accelerate observable effects, it significantly exceeds the typical storage conditions for pharmaceutical. It would be beneficial to conduct a study to ascertain whether analogous degradation processes occur at standard storage temperatures over extended time periods.
A notable constraint pertains to the absence of evaluation employing supplementary APIs. The question of whether the observed degradation is specific to Re or represents a broader phenomenon applicable to other compounds remains unresolved.
The control experiment—Re in aqueous solution—is insufficient to exclude potential non-MCC-related degradation. Inclusion of an additional control with inert material with comparable surface area but minimal surface reactivity, would enhance the robustness of the conclusions.
While differences in the activity among MCC samples are noted, the manuscript does not provide a thorough characterization of their properties. Aside from minor variations in supernatant pH (reported to be between 6.46 and 7.10, but no information provided on whether there is a correlation with catalytic activity), no surface characterization (e.g., characterization of acid−base properties of surface by X-ray photoelectron spectroscopy, solid-state NMR, etc.) is provided.
Data tables present means with confidence intervals; however, the type of statistical measure (standard deviation?) is not specified. Additionally, the number of replicates (n) for each measurement is omitted, which is essential for evaluating the reliability of the results.
Minor comments:
In section 2.2.2. Determination of pH of microcrystalline cellulose, brief information about the method should be provided.
Line 137: Chromolith
Lines 165, 184, 188, Table 2: Should it be micromoles?
There are many spelling mistakes in the supplementary.
Author Response
Comments and Suggestions for Authors
Reviewer 2:
Under the Editor’s requirements, the Abstract has been revised to consist of five sections with headings of Background/Objectives, Methods, Results, and Conclusions. And the structure of the manuscript has been changed to the following sections: Introduction, Results, Discussion, Materials and Methods, and Conclusions.
The subject matter of the article is undoubtedly relevant and merits further discussion. Microcrystalline cellulose (MCC) is a prevalent constituent in pharmaceutical formulations; however, the extent of its potential interactions with active pharmaceutical ingredients (API) remains insufficiently characterised. In the context of increasing regulatory interest in the stability of dosage forms this topic is of clear practical importance. The submitted manuscript is satisfactory in general and appropriate for publication in the journal. Although the experimental data appear to support the hypothesis that MCC may contribute to the degradation of Re under the specified conditions, there are several significant and minor inaccuracies that must be rectified before the manuscript can be considered for publication.
Major Comments:
1. The selection of ginsenoside Re as the model compound is a contentious issue. The structural complexity of Re, in conjunction with the absence of a well-defined and characterised degradation profile, introduces ambiguity. The isolation and structural confirmation of the degradation products was not carried out, which limits the interpretability of the results. A more suitable choice would be a well-characterised, pharmaceutically relevant compound with known degradation pathways.
The degradation pathway of ginsenosides under acidic or alkaline conditions has been well described in the literature [1, 2].
- Zheng MM, Xu FX, Li YJ, Xi XZ, Cui XW, Han CC, et al. Study on Transformation of Ginsenosides in Different Methods. BioMed research international. 2017;2017:8601027.
- Quan K, Liu Q, Wan J-Y, Zhao Y-J, Guo R-Z, Alolga RN, et al. Rapid preparation of rare ginsenosides by acid transformation and their structure-activity relationships against cancer cells. Scientific Reports. 2015 2015/02/26;5(1):8598.
2. The experimental setup involves exposure to elevated temperatures (a water bath at 90°C for 6 hours), which are known to promote the degradation of many natural compounds. While this approach may accelerate observable effects, it significantly exceeds the typical storage conditions for pharmaceutical. It would be beneficial to conduct a study to ascertain whether analogous degradation processes occur at standard storage temperatures over extended time periods.
Yes! The investigation on the hydrolysis of ginsenoside Re at a standard storage temperature should provide further useful information. We did not conduct the experiments at lower temperatures because we focused on whether ginsenoside Re is applicable as a probe to evaluate the MCC catalytic potential of an MCC.
3. A notable constraint pertains to the absence of evaluation employing supplementary APIs. The question of whether the observed degradation is specific to Re or represents a broader phenomenon applicable to other compounds remains unresolved.
To our knowledge, the phenomenon is applicable to all Dammarane-type ginsenosides that contain a 20-glycosyl. However, because the reactant of H+ was immobilized on the cellulose, the reactions are much slower when a reaction is conducted on an MCC due to the space effects.
4. The control experiment—Re in aqueous solution—is insufficient to exclude potential non-MCC-related degradation. Inclusion of an additional control with inert material with comparable surface area but minimal surface reactivity, would enhance the robustness of the conclusions.
We tested the effects of several polyhydroxylated materials, i.e., cotton wool, filter paper, and silicon, on the hydrolysis of ginsenoside Re (Section 3.2 in the previous vision). These materials did not display catalytic effects.
5. While differences in the activity among MCC samples are noted, the manuscript does not provide a thorough characterization of their properties. Aside from minor variations in supernatant pH (reported to be between 6.46 and 7.10, but no information provided on whether there is a correlation with catalytic activity), no surface characterization (e.g., characterization of acid−base properties of surface by X-ray photoelectron spectroscopy, solid-state NMR, etc.) is provided.
No, we did not test the surface of the MCCs. We focus on the acidic catalytic activities in this job, and the possibility of sulfuric acid immobilization that provides an acidic environment on a swelled MCC. We tested the IR spectra of a sulfuric acid-prepared MCC (MCC-S) and cotton wool (Fig. 1). However, the signal is too weak to provide solid evidence for the existence of the sulfonic group on the surface due to the low content.
Fig. 1. IR of self-prepared MCC (MCC-S) and the cotton wool from which the MCC-S was made.
6. Data tables present means with confidence intervals; however, the type of statistical measure (standard deviation?) is not specified. Additionally, the number of replicates (n) for each measurement is omitted, which is essential for evaluating the reliability of the results.
The statistical measure type of the data and the number of replications have been added.
Minor comments:
7. In section 2.2.2. Determination of pH of microcrystalline cellulose, brief information about the method should be provided.
A brief description of the sample preparation procedure for MCC pH estimation has been added.
8. Line 137: Chromolith
Yes, the column used was Chromolith. The text has been corrected.
9. Lines 165, 184, 188, Table 2: Should it be micromoles?
Yes, the concentration of the solution was micromole (mM). The font of the characters is “Symbol”; however, it was changed automatically when the text was copied to the manuscript template.
10. There are many spelling mistakes in the supplementary.
The spelling mistakes in the Supplementary Data have been checked.

Reviewer 3 Report
Comments and Suggestions for Authors
Reviewer’s comments:
The problem discussed in the manuscript, namely the catalytic properties of microcrystalline cellulose (MCC), seems interesting in heuristic chemical sense and important for purposes of pharmacology. Despite of the simplicity of experiments (Figures 1 and 2 together with Supplemental data), the results are quite convincing. The Conclusion (sub-section 4) contains a clear statement: “The pH value of the supernatant of microcrystalline cellulose does not represent the acidity on its surface. The first reason is because there might be immobilized acid that is not so easily disassociated (correct misprinting, should be “dissociated”) into the media, and the second is that hydrogen bonding could make the H3O+ content on the surface richer than that in the supernatant.
Unfortunately, there is typical problem for many articles; it is rounding the numerical data and the number of significant digits in them. Let us consider Table 1: the value 89.23±9.82 is unacceptable. It should be 89 +- 10. The same for other numbers, namely 71.23±9.26 (should be 71 +- 9), 4.45±0.66 (should be 4.4 +- 0.7), etc. Similar problems exist in Table 2: e.g., instead of 96.86±0.68 should be 96.9 +- 0.7. At the same time the value 99.67±0.09 is absolutely correct. The “Calculated pH” data deserve a separate comment: the values 4.5311, 4.8321, and 5.1330 contain FIVE significant digits. The reviewer is not sure that equipment for pH measuring with such high accuracy actually exists.
Similar problems with number of data rounding and number of digits should be corrected in Tables 3 an4, as well. The example from Table 4 is the value 85.95±13.31; after correct rounding it converts into 86 +- 13.
Besides mentioned, the reviewer recommends two small corrections. First is at lines 150-151: do not separate the value (-15) and its unit (V) by transferring to another line. The second – please check the symbol of potential in line 257.
However, all remarks mentioned above not necessary to classify as serious problems. All of them are rather the subjects for minor revision.
Author Response
Comments and Suggestions for Authors
Reviewer 3:
Reviewer’s comments:
Under the Editor’s requirements, the Abstract has been revised to consist of five sections with headings of Background/Objectives, Methods, Results, and Conclusions. And the structure of the manuscript has been changed to the following sections: Introduction, Results, Discussion, and Materials and Methods.
The problem discussed in the manuscript, namely the catalytic properties of microcrystalline cellulose (MCC), seems interesting in heuristic chemical sense and important for purposes of pharmacology. Despite of the simplicity of experiments (Figures 1 and 2 together with Supplemental data), the results are quite convincing. The Conclusion (sub-section 4) contains a clear statement: “The pH value of the supernatant of microcrystalline cellulose does not represent the acidity on its surface. The first reason is because there might be immobilized acid that is not so easily disassociated (correct misprinting, should be “dissociated”) into the media, and the second is that hydrogen bonding could make the H3O+ content on the surface richer than that in the supernatant.
The word “disassociated” has been changed to “dissociated”.
Unfortunately, there is typical problem for many articles; it is rounding the numerical data and the number of significant digits in them. Let us consider Table 1: the value 89.23±9.82 is unacceptable. It should be 89 +- 10. The same for other numbers, namely 71.23±9.26 (should be 71 +- 9), 4.45±0.66 (should be 4.4 +- 0.7), etc. Similar problems exist in Table 2: e.g., instead of 96.86±0.68 should be 96.9 +- 0.7. At the same time the value 99.67±0.09 is absolutely correct. The “Calculated pH” data deserve a separate comment: the values 4.5311, 4.8321, and 5.1330 contain FIVE significant digits. The reviewer is not sure that equipment for pH measuring with such high accuracy actually exists.
The significant digits of the data in the tables have been revised.
Similar problems with number of data rounding and number of digits should be corrected in Tables 3 an4, as well. The example from Table 4 is the value 85.95±13.31; after correct rounding it converts into 86 +- 13.
The significant digits of the data in the tables have been revised.
Besides mentioned, the reviewer recommends two small corrections. First is at lines 150-151: do not separate the value (-15) and its unit (V) by transferring to another line. The second – please check the symbol of potential in line 257.
The separates were removed. The symbol in line 257 is because the font type was changed automatically when the text was rearranged according to the template. The correct font is “Symbol”. Similar errors at lines 69, 71, 165, 184, and 188 have also been corrected.
However, all remarks mentioned above not necessary to classify as serious problems. All of them are rather the subjects for minor revision.
Thank you for your positive comments and constructive suggestions.
Round 2
Reviewer 2 Report
Comments and Suggestions for Authors
The revised manuscript has been modified to address all critical comments and is now suitable for publication in the journal.